# Interleukin 17 and Its Involvement in Renal Cell Carcinoma

**DOI:** 10.3390/jcm11174973

**Published:** 2022-08-24

**Authors:** Michał Jarocki, Julia Karska, Szymon Kowalski, Paweł Kiełb, Łukasz Nowak, Wojciech Krajewski, Jolanta Saczko, Julita Kulbacka, Tomasz Szydełko, Bartosz Małkiewicz

**Affiliations:** 1University Center of Excellence in Urology, Department of Minimally Invasive and Robotic Urology, Wroclaw Medical University, 50-556 Wroclaw, Poland; 2Department of Molecular and Cellular Biology, Faculty of Pharmacy, Wroclaw Medical University, 50-556 Wroclaw, Poland

**Keywords:** interleukin-17, renal cell carcinoma, immunotherapy, inflammation, tumor microenvironment, tumor development, Th17 lymphocytes

## Abstract

Nowadays, molecular and immunological research is essential for the better understanding of tumor cells pathophysiology. The increasing number of neoplasms has been taken under ‘the molecular magnifying glass’ and, therefore, it is possible to discover complex relationships between the cytophysiology and immune system action. An example could be renal cell carcinoma (RCC) which has deep interactions with immune mediators such as Interleukin 17 (IL-17)—an inflammatory cytokine reacting to tissue damage and external pathogens. RCC is one of the most fatal urological cancers because of its often late diagnosis and poor susceptibility to therapies. IL-17 and its relationship with tumors is extremely complex and constitutes a recent topic for numerous studies. What is worth highlighting is IL-17’s dual character in cancer development—it could be pro- as well as anti-tumorigenic. The aim of this review is to summarize the newest data considering multiple connections between IL-17 and RCC.

## 1. Introduction

IL-17 is a signature cytokine of T helper 17 (Th17) cells. Its role in the autoimmunity, host defense against pathogens, and allergic diseases are well studied [1]. However, over the years, it became apparent that IL-17 is involved in far more physiological and pathological conditions than was initially suggested [2]. One of more recent directions of IL-17’s research is its relationship with cancer. Many studies indicate that IL-17 could be, at least partially, responsible for the creation of a cancer-promoting environment [3]. In this regard, its relation to renal cell carcinoma (RCC) seems to be no different. RCC is among the 10 most common cancers worldwide and a heavy burden on healthcare [4]. Thanks to evolving immunopathological techniques and better understanding of IL-17’s pathways, the link between RCC and IL-17 is more evident than ever. The aim of this review is to summarize current understanding of IL-17’s role in RCC development and management.

## 2. Evidence Acquisition

For the purposes of this narrative review, we conducted a comprehensive English language literature research for original and review articles using the PUBMED/Scopus database through May 2022. We searched for the combination of following terms: interleukin-17; Th17 lymphocytes; renal cell carcinoma; tumor microenvironment. We found 770 related articles, and the final number of papers that were selected for this manuscript was 86. Studies with the highest level of evidence and relevance to the discussed topics (32) were selected, with the consensus of the authors.

## 3. RCC—A Brief Overview

Renal cell carcinoma is a heterogenous group of cancers originating from the renal cortex. They constitute approximately 85% of all primary renal neoplasms, with transitional cell carcinomas, oncocytomas, angiomyolipomas, and renal sarcomas occurring less frequently [5]. Although RCCs account only for 2% of all cancers, their fatality rate is notoriously high, and when metastasis occurs, the 5-year survival rates are notably low [6]. These properties can be attributed to the usually late diagnosis of renal tumor (due to it being asymptomatic in early stages) and poor susceptibility to systemic chemotherapy and radiotherapy. Most RCCs are discovered accidentally on imaging and prognosis varies heavily depending on the stage of disease. While small and localized tumors have a high 5-year relative survival rate at 93%, it drops down significantly when metastases develop to only 12% [6]. Due to the fact that about a third of RCCs are diagnosed at Stage IV (metastatic) and 20–50% of pre-metastatic cases will progress to Stage IV regardless of surgical excision, the RCC is considered the deadliest urological cancer [7]. The most common type of RCC at 75% occurrence rate is clear cell RCC (ccRCC), followed by papillary RCC (16%) and chromophobe RCC (7%). Overall, the 5-year survival rate of RCC is around 75%.

## 4. IL-17—A Versatile Lymphokine

IL-17A (traditionally known as IL-17, in the past also termed as CTLA-8) is the first and best characterized member in its family composed of IL-17A, B, C, D, E (also known as Il-25), and F [8]. IL-17 was first discovered in 1993 [9], although it became well-known in 2005 with the finding of a new population of CD4+ Th cells—Th17 that expressed this cytokine [10]. Naive CD4+ T-cells are triggered to differentiate into Th17 in the presence of both TGF-β and Il-6. The differentiation is characterized by the production of IL-17, Il-21, and ROR-yt (transcription factor). The differentiation of Th17 cells also depends on dendritic cells which produce IL-1, IL-6, and IL-23. These molecules preferentially activate STAT-3 which induces transcription factor ROR-yt. ROR-yt is also expressed in the presence of STAT-3 that is activated by Il-6, Il-21, and Il-23. There is also an autocrine generation of Th17 by Il-21 that is derived from these T-cells. Moreover, Il-21 leads to Il-23 receptor expression of Th17 and their susceptibility to Il-23 (produced by dendritic cells) stimulation. This cytokine gives Th17 phenotype stability and helps them with acquiring effector functions [11].

There are also other cells that have been discovered that secrete IL-17 such as γδ+ T cells, NK T cells, and TCRβ+ natural Th17 cells and Type 3 innate lymphoid cells (ILC3) [12]. The family of IL-17 receptors (IL-17R) comprises of subunits: IL-17RA, IL-17RB, IL-17RC, IL-17RD, and IL-17RE [13]. IL-17 and IL-17F occur in a homodimer or as a heterodimer and induce signals affecting dimeric IL-17RA and IL-17RC receptor complex [8]. IL-17R has a single transmembrane domain with a long cytoplasmic tail which indicates multiple regulatory domains triggering in the presence of receptor signaling diverse functions. In this process, the adaptor protein Act1 joins the receptor subunits and activates cell signaling pathways by different TNF receptor–associated factor (TRAF) proteins which activates many transcription factors [8]. This IL-17R–Act1 complex cooperates as well with MEKK3 and MEK5 in a TRAF4-dependent manner that results in the activation of ERK5 [8]. IL-17R are expressed mainly by non-hematopoietic cells, but their presence is generally ubiquitous [11].

IL-17 is known as a proinflammatory interleukin that reacts by a tissue damage or a pathological situation that is caused by an external pathogen such as fungi or extracellular bacteria or by cancer. These functions are possible through gene expression—driving de novo gene transcription or stabilizing target mRNA transcripts. There is a unique IL-17 gene signature” that consists of cytokines (IL-6, IL-1, G-CSF, GM-CSF, and TNF-α); chemokines (CXCL1, CXCL2, CXCL5, CCL2, CCL7, and CCL20); antimicrobial peptides (β-defensins, S100 proteins, and lipocalin2); and matrix metalloproteinases (MMPs) (MMPs 1, 3, 9, and 13) [8,13]. The gene target although depends on the organ tissue, e.g., in the kidney that is infected by *Candida albicans,* IL-17 upregulates protective genes of the kallikrein–kinin system (KKS) [14]. Transcription factors that are activated by IL-17 are NF-κB and C/EBP, whereas IL-17-induced pathways are MAPK ones with ERK, p38, and JNK [14,15]. These processes lead to the expansion and accumulation of neutrophils in the innate immune system [16], modulate the interplay between commensal microbes and epithelial cells at skin or mucosae) preserving its integrity [17], and play role in the pathogenesis of allergies, autoimmune diseases (e.g., rheumatoid arthritis, psoriasis, inflammatory bowel disease), allograft transplantation, and even malignancy. However, IL-17 supports host defense against bacterial (e.g., extracellular *Klebsiella pneumoniae*, intracellular *Listeria monocytogenes*), mycobacterial, and fungal (especially *Candida albicans*) pathogens [18] and also some types of cancer [16].

## 5. IL-17 in Kidney Diseases

While the direct mechanism of IL-17 interaction in RCC remains elusive, we can safely extrapolate available data regarding IL-17 role regarding cancer in general. IL-17 is associated with chronic inflammation in many diseases and kidney diseases are no different. Increased levels of IL-17 are found in autoimmune kidney diseases and the serum level of IL-17 strictly correlates with disease activity [19]. The IL-17 receptors are expressed in many kidney cells (podocytes, mesangial, and renal endothelial cells) and are responsible for the stimulation of the inflammatory environment which leads to the damage and disturbance of nephron function [20]. Additionally, IL-17 receptors stimulate profibrotic pathways causing fibrosis and eventually the loss of organ function [21]. The result of these interactions is that autoimmune kidney diseases, when treated poorly or insufficiently, may lead to the onset of chronic kidney disease (CKD) which, in turn via various pathways, increases the risk of cancer development [22].

## 6. IL-17 in Tumors

IL-17 plays a dual role in tumor development. It is worth noting that IL-17 may have both stimulating and inhibiting effects on the growth of tumor cells. The potential role of IL-17 includes tumorigenesis, proliferation, angiogenesis, and metastasis [23,24]. IL-17 stimulates the secretion of cytokines such as G-CSF which recruit myeloid-derived suppressor cells (MDSC). MDSCs are immature myeloid cells with the ability to decrease the adaptive immunity [25,26]. These cells are recruited into the neoplastic tissue via IL-17-induced chemokines (CXCL1/CXCL5). IL-17 also induces pro-inflammatory NF-kB and Il-6, which acts as paracrine signalers and increases tumor growth and survival. Recent studies have shown that IL-17, independently of its effect on the tumor microenvironment, directly induces the proliferation of precancerous cells. An association has also been demonstrated between IL-17 signaling and tumor formation within wounds during healing. This is due to the induction of Lrig1 + cells with oncogenic mutations such as KrasG12D [25].

In summary, in the early stages of tumor development, IL-17 acts in a multidirectional way: (1) by activating MDSC, which results in decreasing systemic immunity, (2) increasing local inflammation, and (3) directly affecting tumor cells. Research has shown both pro- and anti-angiogenic effects. The angiogenic and lymphangiogenic effects are observed via the activation of VEGF. As a result, it promotes the formation of metastases [24,25,26]. IL-17s influence on early tumor development is summarized in Figure 1.

However, studies have indicated that the IL-17 knockout contributes to a more aggressive metastatic picture, which may be related to the loss of the potent antitumor cytokine IFN-gamma [24]. It has been shown that IL-17A directly inhibits the growth of P815 mastocytoma and J558L plasmacytoma [27,28]. There is also known evidence that IL-17A may exert an indirect pro-apoptotic effect. It was found that IL-17A, by increasing the production of inducible nitric oxide synthase (iNOS), leads to an increase in the synthesis of nitric oxide (NO). It may be an important factor, which can inhibit the growth of oral squamous cell carcinoma cells via apoptosis [29]. The tumor cell growth inhibitory effect was also observed with IL-17E. IL-17E in combination with chemotherapeutic agents inhibits the growth of melanoma and pancreatic tumors in mice [30].

Interactions between the microbiota and IL-17A-producing cells are also involved in the pathogenesis of immune-mediated inflammatory diseases and cancer. Studies have shown that the gut microbiome can affect the activity of Th17 cells and IL-17A that is induced by the microbiota is also associated with the pathogenesis of colon cancer, breast cancer, pancreatic cancer, ovarian cancer, and multiple myeloma [17].

## 7. IL-17s Role in Carcinogenesis

### 7.1. Protumor

There is little data about the positive influence of IL-17 on RCC. It is known that Th17 lymphocytes that produce IL-17 are said to be higher concentrated in the serum of RCC patients [31]. Moreover, transcription factor typical for Th17—RORC is upregulated in this group. RORC is responsible for directing the Th17 lineage which suggests that this correlation leading to abnormal differentiation of lymphocytes may play an important role in both the occurrence and progression of RCC [31,32]. Furthermore, with the higher tumor stage and grade, the number of Th17 increases. High concentration of these lymphocytes is also connected with a decreased survival rate. Taking in consideration this evidence, Th17 cells could be clinical markers of RCC diagnosis and progression together with patients survival rate [31].

IL-17 has also a bidirectional influence on RCC cells [33]. Inzoume et al., isolated T-cell clones from one patient before any systemic treatment and generated them by in vitro stimulation with dendritic cells, autologous tumor, and IL-2 in the presence of anti-CTLA4 antibody. In this environment of a.o. RCC cells, T-lymphocytes were triggered to produce IL-17 that consequently induces tumor cells to release large amounts of Il-8. IL-17 was already known to be responsible for inflammatory cytokine production from pulmonary epithelium or smooth muscle [34] and even IL-8 from tumor cells, but not on such a high level [35]. What is important, is that IL-8 in turn has a chemotactive influence on T-cells and angiogenesis which leads to RCC local infiltration by T-lymphocytes, however it may not be a single factor in this phenomenon [33]. The interplay between IL-17 and RCC is summarized in Figure 2.

Interleukin 17 is a pro-inflammatory cytokine that promotes chemotaxis and degranulation of neutrophils. The increased expression of IL-17 has been demonstrated in the tumor microenvironment [26]. Research suggests a relationship between IL-17 production and circulating MDSC levels [36]. MDSCs exert an influence on the adaptive immune response and regulate innate immunity by modulating cytokine production by macrophages [37]. It has also been reported the role of MDSC in angiogenesis and tumor promotion [38]. However, the relationship between cytokine production and MDSC accumulation in RCC is not fully understood. In addition to IL-17, IL-18 and CCL2 also play an important role in this phenomenon. Guan et al., showed that in addition to IL-6 and tumor necrosis factor alpha (TNF-α), IL-17 has been shown to be overexpressed in RCC tissue, and its expression is dependent on the degree of malignancy. They also described that parenchymal levels of IL-17 correlate with increased total MDSC [26].

IL-17 also influences many other cancers, and these phenomena can indicate further research on IL-17 and tumorigenesis of RCC. IL-17B supports pancreatic cancer by up-regulating ERK pathways propagating tumor cell invasion and facilitates metastatic cell survival. Consequently, the overexpression of IL-17B receptors that are found on these cells coincided with poorer prognosis for patients [39]. Furthermore, IL-17 was found to negatively impact colorectal cancer cells by promoting oncogenesis [40]. Altogether, these discoveries implicate the immense role of IL-17 in cancer development on many of its phases, from oncogenesis to metastases.

The interplay between T-cells, IL-17, and RCC environment is promising for a better understanding of this cancer, but it is also complicated and requires further research in this topic.

### 7.2. Prometastatic

Chronic inflammation contributes to tumor metastasis via an unknown pathway. Probably, tumor hypoxia accompanying rapid growth triggers angiogenesis that helps in metastasizing. However, not only nutrients flow through new vessels, but also immune cells to tumor mass [11]. Taking into consideration this dual nature of angiogenesis and inflammation effect on tumor, the role of IL-17 in it has been investigated.

IL-17 is known for both its pro-tumor and anti-tumor effects [11,41]. When supporting a tumor and its metastasis, IL-17 induces angiogenesis and sustains an inflammatory environment. Firstly, IL-17 increases the production of VEGF, which consequently triggers TGF-β- and VEGF-mediated angiogenesis. In turn, TGF upregulates VEGF receptor expression enhancing VEGF receptivity [11]. Secondly, the production of IL-8 is stimulated by IL-17. IL-8 leads to angiogenic responses in endothelial cells and, therefore, increases proliferation and survival of both endothelial and cancer cells, potentiating the migration of cancer cells [11,35]. Thirdly, the production of prostaglandin E1 and prostaglandin E2 which stimulate tumor angiogenesis is enhanced by IL-17 [42]. Finally, IL-17 increases the production of IL-6 in fibroblasts and Il-6 induces Th17 differentiation and, therefore, IL-17 expression. This creates a chronic inflammatory state that supports tumor growth and metastasis [11]. The influence of IL-17 on tumorigenesis is summarized in Figure 3.

In osteosarcoma and synovial sarcoma, the positive influence of IL-17 on metastasizing has been especially evidenced [43,44]. In the first one, IL-17 through activating its receptor IL-17R upregulates the expression of VEGF and CXCR4, which are known to take part in the metastasis of tumors [43,45,46]. Moreover, Stat3 that induces tumor progression via supporting tumor survival, angiogenesis, and suppressing antitumor immunity is activated by IL-17 [43,47,48]. What is similar in the pro-metastasis role of IL-17 between osteosarcoma and synovial sarcoma, is MMP (matrix metalloproteinase) [43,44]. IL-17, through IL-17R, enhances the expression of MMP, which is required for the proteolytic modifications of basement membranes and extracellular matrices by angiogenesis and metastasis [43,44,49,50,51]. Additionally, in synovial sarcoma, metastasizing is supported by IL-17 through the expression of VEGF and CXCR4 [44].

### 7.3. Antitumor

Even though IL-17 can be considered a tumor-promoting factor in many cases, research shows that its role is far more complex. IL-17 expression in RCC and also other cancers seems to correspond with an overall better prognosis. Huang et al., identified that the patients with ccRCC that was enriched with VHL, but depleted of BAP1 mutations and with high levels of Th17 and CD8+ T-cells were characterized by longer survival [52].

In some other human cancers, it has also been shown that the infiltrating level of Th17 cells in tumor mass was negatively correlated with poorer survival, tumor growth, and stage [53]. A study on gastric adenocarcinoma patients revealed that high intratumoral expression of IL-17 results in significantly higher five-year overall survival. Even though the exact role of IL-17 remains to be studied, patients with a greater expression of IL-17 had the likelihood of death at 5 years reduced by a staggering 48% [54]. Furthermore, research that was conducted on cervical adenocarcinoma patients also supports the anti-tumor properties of IL-17 theory. An increased levels of IL-17 and IL-17(+) cells were found in patients with smaller tumor sizes, less infiltration depth, and the absence of vasal invasion [55]. Additionally, in patients with chronic lymphocytic leukemia increased Th17 and IL-17 amounts corresponded with a longer overall survival rate [56]. Therefore, it is essential to view IL-17 as an important factor in tumor immunity and research further its complex role, as similar pathways and effects could occur in RCC.

### 7.4. Antimetastatic

Some studies indicate that IL-17 can indirectly inhibit tumor metastasis—here IL-23 plays the main role. On the one hand, it triggers Th17 differentiation, IL-17 production, and, therefore, maintains inflammatory environment, favorable for metastasis. On the other hand, IL-23 may also have anti-tumor effects, because tumors that overexpressed this cytokine presented reduced growth and metastasis [11,57,58,59,60].

## 8. IL-17 in RCC Detection

Currently, more attention is being paid to the role of biomarkers in the diagnosis and prognosis of neoplastic diseases [61,62]. Diagnostic biomarkers can allow for an early detection and classification of cancer. Prognostic biomarkers can inform clinicians about the natural course of an individual cancer and guide their decision of whom to treat and how intensive the treatment should be [61,62]. In the case of RCC, which is highly diverse in the molecular field, the latest advances in proteomics, genomics, and metabolomics could be particularly useful [61]. The latest literature describes a number of substances and molecules that could constitute these markers, which can be used in the early detection of RCC, providing the benefits of new technologies [62]. The authors indicate immunological markers and immune checkpoint inhibitors as one of the important biomarkers [61]. Due to its direct involvement in RCC pathogenesis, IL-17 may also be considered as one of them. However, none of these biomarkers have yet been established for the routine clinical use in management of this cancer and more research seems to be necessary [61].

## 9. RCC Treatment

Currently, surgical resection is the mainstay of treatment for renal cell carcinoma [63,64]. This method is curative in patients with localized disease [65]. We can distinguish radical nephrectomy, which is the gold standard for T1b-T4 cancers, and nephron-sparing surgery that is used for small tumors and showing the same effectiveness compared to total nephrectomy [66]. In the case of primary disseminated disease, cytoreductive nephrectomy with metastasectomy is combined with targeted therapy. However, currently, in the era of targeted therapies, the role of these surgeries is not precisely defined and further research is needed [67].

Open simple enucleation is an alternative to classical partial nephrectomy for 4–7 cm renal tumors. It provides long-term cancer-specific survival rates that are similar to those of radical nephrectomy [68]. The utility and efficacy of minimally invasive techniques were assessed in the ERASE (endoscopic robot-assisted simple enucleation) study. Mari et al., showed that it provides excellent oncological results [69].

Some kidney tumors have the unique feature of growing into the venous system, which may be a complex surgical challenge [66]. Studies have shown that up to 10% of RCC patients have tumors that infiltrate the venous system [70]. Recent studies have initially shown that minimally invasive surgery provides favorable perioperative outcomes in patients with RCC and inferior vena cava (IVC) thrombosis. However, there is a need for further research within this topic [71]. Regrettably, 25% to 30% of patients have distant metastatic disease at the time of diagnosis and 40% of patients eventually, after surgical treatment, develop recurrence [63]. Importantly, the management of mRCC requires a multidisciplinary approach and aiming the optimal treatment strategy to patients with RCC [72]. Formerly, tyrosine kinase inhibitors (TKI) were the only drugs for the first-line treatment of metastatic renal cell carcinoma (mRCC) [73]. Recently approved immune checkpoint inhibitors (such as PD-1-blocker nivolumab) have changed the standard of metastatic kidney tumor therapy and the use of immune checkpoint inhibitors in mRCC presents a new age of treatment of this disease [63,73].

Future strategies for advanced renal cell carcinoma include new kinase inhibitors, novel combinations, the use of new checkpoint inhibitors, and targeted therapies. It has also been suggested that IL-17 as an immune checkpoint may be an interesting therapeutic target [23].

## 10. Drugs Targeting IL-17 Axis

Due to the involvement of IL-17 in carcinogenesis and its role in tumor resistance to treatment, many existing drugs that are known for their application in autoimmune diseases therapy could have a potential role in general oncological chemotherapy. There is, however, a lack of evidence supporting their application in RCC. The application of such drugs could have effects that are not easily foreseeable due to the dualistic (protumor vs. protective) nature of IL-17’s influence on RCC. Despite limited literature on the subject and the fact that currently no trials evaluating the effect of IL-17-interacting drugs on RCC are present, their application, at least theoretically, could be beneficial for patients. Therefore, further research needs to be done regarding the effects of interfering with the IL-17 axis in RCC patients. Currently, drugs targeting the IL-17 axis are used mainly as a second and third line treatment in some autoimmune diseases and include Secukinumab, Brodalumab, Ixekizumab, and MSB0010841/ALX-0761. The mechanism of action of these drugs, as well as their current therapeutic applications are summarized in Table 1.

### 10.1. Secukinumab

Secukinumab is an anti-interleukin-17A IgG1 monoclonal antibody [74] with potential application in a large variety of diseases. The FDA approved it for the treatment of severe psoriasis, some psoriasis-related diseases, and ankylosing spondylitis. It can also be used off-label in the treatment of SLE (systemic lupus erythematosus) and RA (rheumatoid arthritis) [75]. Even though the safety profile of secukinumab is favorable, some research indicates the higher risk of the development of cancer in secukinumab-treated patients [76] while others suggest that the risk is not increased compared to the reference population [77]. Although no research has evaluated the effect of secukinumab on cancer per se, it is important to further research its interactions and potential applications in this regard.

### 10.2. Brodalumab

Brodalumab is a fully human IgG2 antibody which binds to IL-17 receptor A that is approved for the treatment of chronic plaque psoriasis in North America and Europe [78]. Interestingly, due to the different mechanism of action, it was effective in patients where treatment with Secukinumab and Ixekizumab failed [79]. In research evaluating malignancy rates in brodalumab-treated patients, it was concluded that the risk of developing cancer was generally low, and indicated limited clinical evidence of brodalumab-related malignancies [80].

### 10.3. Ixekizumab

Ixekizumab is a humanized IgG4 anti-IL-17A antibody neutralizing circulating IL-17A that is approved for the treatment of ankylosing spondylitis [81]. Its use in plaque psoriasis resulted in a 75% reduction in the psoriasis area severity index (PASI 75) [82]. A case study on patients with psoriasis and laryngeal cancer indicated the possible beneficial effect of ixekizumab on laryngeal cancer [83]. This claim is supported by research suggesting the increased apoptosis of LC cells when IL-17 is inhibited [84].

### 10.4. MSB0010841/ALX-0761

MSB0010841/ALX-0761 is a trivalent nanobody that is specific to both IL-17A and IL-17F [85]. Its application in psoriasis and refractory/relapsed B-cell lymphoma is currently being evaluated in clinical trials (Phase II and I respectively) [86].

Drugs targeting the IL-17 axis are wide-spread and effective in the treatment of some autoimmune diseases. There is, however, a lack of research regarding their potential application in cancer. Due to the fact that many publications indicate the profound role of IL-17 in carcinogenesis, the urge for further research of IL-17-interacting drugs and their application needs to be emphasized.

## 11. Conclusions

IL-17 is an important factor that is vital in both physiological and pathological processes. Recent studies indicate its immense role in every step of tumor development. Its involvement in cancer can be traced from the complicity in the creation of a tumor-promoting environment to the metastatic stage. Therefore, many authors consider it a pro-tumoric factor, as its best documented contributions to cancer include tumorigenesis, proliferation, angiogenesis, and metastasis. In RCC, its role seems to be primarily pro-tumoric as well, although its potential antineoplastic role requires further research. This tumor-inhibiting property is reinforced by research on different malignancies in which the activity of IL-17 was associated with better prognosis. Thus, it seems that IL-17’s involvement in tumors is more of a dualistic, “double-edged sword” nature. RCC is a deadly and common type of cancer that is responsible for over 130,000 deaths yearly, worldwide. A better understanding of IL-17’s immuno-oncological impact, as well as the potential application of drugs interfering with IL-17 axis may significantly improve future RCC management strategies.

## Figures and Tables

**Figure 1 jcm-11-04973-f001:**
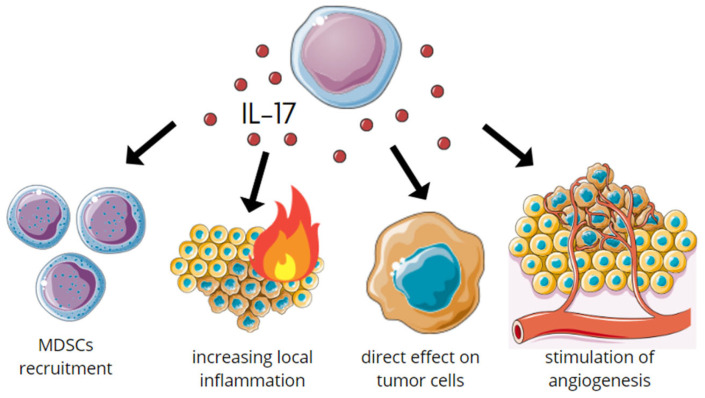
Multidirectional action of IL-17 in the early stages of tumorigenesis.

**Figure 2 jcm-11-04973-f002:**
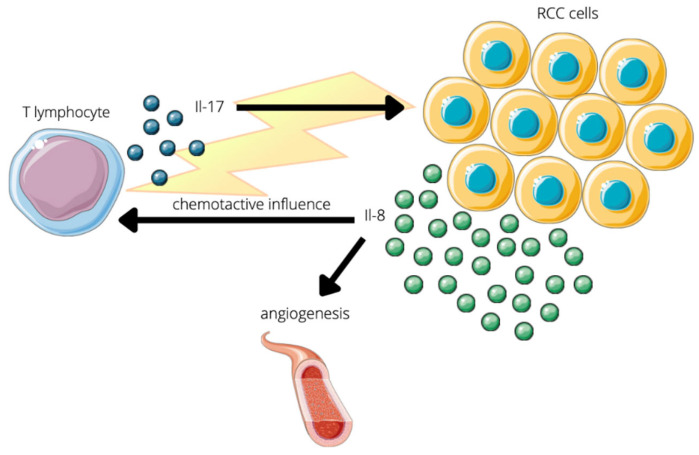
Schematic overview of the bidirectional influence between T-lymphocytes and RCC. (1) T-lymphocytes that were isolated from the RCC patient and incubated in special conditions, produce Il-17 in the presence of RCC cells. Il-17 triggers tumor cells to release a large amount of Il-8. This cytokine induces not only angiogenesis, but also has a chemotactive impact on T-lymphocytes. (2) Consequently, the high number of lymphocytes T infiltrate RCC and tumors new vessels are created which is a favourable environment for tumor development.

**Figure 3 jcm-11-04973-f003:**
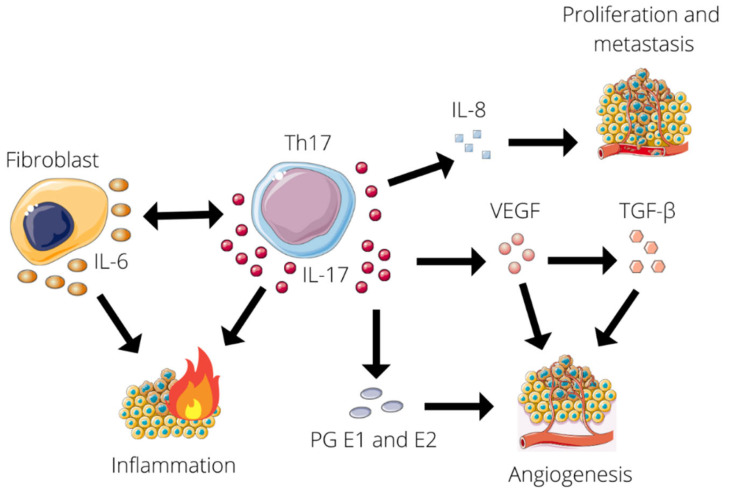
Schematic overview of IL-17’s influence on tumorigenesis. IL-17 stimulates the production of VEGF, PG E1, and E2, IL-6 and IL-8. VEGF triggers VEGF- and TGF-β-mediated angiogenesis, potentiated by the action of PG E1 and E2. IL-6 induces Th17 differentiation and IL-17 expression, stimulating chronic inflammation. IL-8 increases the proliferation and migration of cancer cells.

**Table 1 jcm-11-04973-t001:** Summarized overview of the mechanism of action of drugs targeting IL-17 axis and their current application in therapy.

Drug	Mechanism of Action	Application
Secukinumab	Anti-interleukin-17A IgG1 monoclonal antibody	Severe psoriasis, ankylosing spondylitis, SLE, RA
Brodalumab	Anti-interleukin-17A receptor human IgG2 antibody	Chronic plaque psoriasis
Ixekizumab	Anti-interleukin-17A humanized IgG4 antibody	Ankylosing spondylitis, plaque psoriasis
MSB0010841/ALX-0761	Anti-interleukin-17A and IL-17F trivalent nanobody	Clinical trials in psoriasis and refractory/relapsed B-cell lymphoma

## Data Availability

Not applicable.

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
