# Peer review of "Interleukin 17 and Its Involvement in Renal Cell Carcinoma"

_jcm, 2022, doi:10.3390/jcm11174973_

Round 1

Reviewer 1 Report

In this paper the author analyses role of interleukin 17 in beginning and progression of renal cell carcinoma.

The excursus about the role of interleukin 17 on pathways of development of oncologic disease and its dualistic nature, protumor and pro-metastatic on one hand, protective in term of oncologic aggressiveness on the other hand, like in cases of gastric or cervical adenocarcinoma, seems very interesting and full of future perspectives.

Looking at RCC, is well known chemotherapy resistance; so surgery rest the first line treatment for non metastatic disease, feasible whit mini-invasive technique in different states of disease.

In case of metastatic disease, treatment comprehend different option, included cytoreductive nephrectomy, metastasectomy and adjuvant therapy, witch included TKI, Monoclonal VEGF antibody, mTOR inhibitor and PD-1 inhibitor.

The review seems well executed but I would recommend reconsider the present manuscript after minor revision.

-        When the author write an overview on RCC, could be useful remind that surgical treatment remains gold standard for localized cancer, cite some of the multiple paper existent in literature (per ex. Oncologic outcomes in patients treated with endoscopic robot assisted simple enucleation (ERASE) for renal cell carcinoma: Results from a tertiary referral center; Techniques and outcomes of minimally-invasive surgery for nonmetastatic renal cell carcinoma with inferior vena cava thrombosis: a systematic review of the literature; Simple Enucleation for the Treatment of Renal Cell Carcinoma Between 4 and 7 cm in Greatest Dimension: Progression and Long-Term Survival)

-        As specified from the author itself, interleukin 17 inhibitor drugs are mainly used for second and third line treatment in autoimmune disease. In this vision, the implications of this cytokine in treatment of cancer seems to be weak, as no trial on tumor disease and interleukin 17 inhibitor as already started. So seems necessary more studies, considering even the dualism (protumor vs protective) nature of this cytokine emerged from this paper, before thinking about it in terms of target for treatment and add new findings at the actual state of scientific literature. 

Author Response

Response letter to the Reviewer #1 Report 

We thank the Reviewer for encouraging feedback and appreciate the insightful comments and suggestions.

Below, we provide a point-by-point response to each of the reviewer’s comments.

All changes in the manuscript were highlighted in yellow for clarity

We hope that introduced revisions significantly improve the quality of this review and qualify it for further editorial stages.

Sincerely,

Authors

Major aspects

  1. When the author write an overview on RCC, could be useful remind that surgical treatment remains gold standard for localized cancer, cite some of the multiple paper existent in literature (per ex. Oncologic outcomes in patients treated with endoscopic robot assisted simple enucleation (ERASE) for renal cell carcinoma: Results from a tertiary referral center; Techniques and outcomes of minimally-invasive surgery for nonmetastatic renal cell carcinoma with inferior vena cava thrombosis: a systematic review of the literature; Simple Enucleation for the Treatment of Renal Cell Carcinoma Between 4 and 7 cm in Greatest Dimension: Progression and Long-Term Survival)

Response:

Thank you for this valuable suggestion. We have added a paragraph on surgical treatment and highlighted that it is the standard treatment for localized cancers. According to suggestion, we also took into account the results of research on the use of minimally invasive techniques 

  1. As specified from the author itself, interleukin 17 inhibitor drugs are mainly used for second and third line treatment in autoimmune disease. In this vision, the implications of this cytokine in treatment of cancer seems to be weak, as no trial on tumor disease and interleukin 17 inhibitor as already started. So seems necessary more studies, considering even the dualism (protumor vs protective) nature of this cytokine emerged from this paper, before thinking about it in terms of target for treatment and add new findings at the actual state of scientific literature. 

Response:

Thank you for the comment. We have added a paragraph in “Drugs Trageting IL17 Axis” heading stating that the application of IL-17 targeting drugs in RCC is only theoretical, as no studies or trails justifying it exist. Additionally we highlighted the issues that such application could provoke – the effect could be hard to foresee due to the dualistic nature of IL-17’s interaction with RCC. Furthermore we underlined that IL-17 targeting drugs are currently used as second/third line treatment in limited number of diseases, i.e. some autoimmune diseases.

Reviewer 2 Report

The authors presented a detailed review of the relationships between cytophysiology and immune system in the context of IL-17 and RCC.  The authors provided a well structured review. Overall interesting manuscript that would be of great interest among different groups of readers. Congratulations to the authors.

I have some minor comments before acceptance.

It would better to provide an additive paragraph "Methods" presenting the research strategy, the time frame of interest and what source of data was interrogated (PUBMED Medline, Scopus).

Finally, considering the novel emerging biomarkers in the context of RCC the authors would consider to discuss the recent ICUD-SIU consultation (doi: 10.48083/XLQZ8269; doi: 10.48083/TNGM4076). How the findings presented in this review could complement the clinical daily practice in the RCC scenario?

Author Response

Response letter to the Reviewer #2 Report

We thank the Reviewer for encouraging feedback and appreciate the insightful comments and suggestions.

Below, we provide a point-by-point response to each of the reviewer’s comments.

All changes in the manuscript were highlighted in yellow for clarity.

We hope that the introduced revisions significantly improve the quality of this review and qualify it for further editorial stages.

Sincerely,

Authors

  1. It would better to provide an additive paragraph "Methods" presenting the research strategy, the time frame of interest and what source of data was interrogated (PUBMED Medline, Scopus).

Response: Thank you for this valuable suggestion. Absolutely valid point. We have added a relevant paragraph (2. Evidence acquisition) describing the methodology, terms of searching for articles and search engines. We hope that thanks to this, our review will be more comprehensible from the technical point of view.

  1. Finally, considering the novel emerging biomarkers in the context of RCC the authors would consider to discuss the recent ICUD-SIU consultation (doi: 10.48083/XLQZ8269; doi: 10.48083/TNGM4076). How the findings presented in this review could complement the clinical daily practice in the RCC scenario?

Response: Thank you very much for this valuable comment. We considered the mentioned articles in the context of our review and took advantage of the valuable information we found there. We have added a paragraph “IL-17 in RCC Detection” where we describe the role of novel biomarkers.